# Application of Nanotechnology to Improve the Performance of Biodegradable Biopolymer-Based Packaging Materials

**DOI:** 10.3390/polym13244399

**Published:** 2021-12-15

**Authors:** Arezou Khezerlou, Milad Tavassoli, Mahmood Alizadeh Sani, Keyhan Mohammadi, Ali Ehsani, David Julian McClements

**Affiliations:** 1Department of Food Science and Technology, Faculty of Nutrition and Food Sciences, Tabriz University of Medical Sciences, Tabriz 5166614711, Iran; arezou.khezerlou@gmail.com (A.K.); mtavassoli2006@gmail.com (M.T.); 2Food Safety and Hygiene Division, School of Public Health, Tehran University of Medical Sciences, Tehran 1417614411, Iran; saniam7670@gmail.com; 3Department of Clinical Pharmacy, Faculty of Pharmacy, Tehran University of Medical Sciences, Tehran 1417614411, Iran; keyhanmohammadi72@yahoo.com; 4Department of Food Science, University of Massachusetts Amherst, Amherst, MA 01003, USA; 5Department of Food Science & Bioengineering, Zhejiang Gongshang University, 18 Xuezheng Street, Hangzhou 310018, China

**Keywords:** bioactive compounds, active packaging, nanoencapsulation, controlled release, sustainability, plant-based delivery systems

## Abstract

There is great interest in developing biodegradable biopolymer-based packaging materials whose functional performance is enhanced by incorporating active compounds into them, such as light blockers, plasticizers, crosslinkers, diffusion blockers, antimicrobials, antioxidants, and sensors. However, many of these compounds are volatile, chemically unstable, water-insoluble, matrix incompatible, or have adverse effects on film properties, which makes them difficult to directly incorporate into the packaging materials. These challenges can often be overcome by encapsulating the active compounds within food-grade nanoparticles, which are then introduced into the packaging materials. The presence of these nanoencapsulated active compounds in biopolymer-based coatings or films can greatly improve their functional performance. For example, anthocyanins can be used as light-blockers to retard oxidation reactions, or they can be used as pH/gas/temperature sensors to produce smart indicators to monitor the freshness of packaged foods. Encapsulated botanical extracts (like essential oils) can be used to increase the shelf life of foods due to their antimicrobial and antioxidant activities. The resistance of packaging materials to external factors can be improved by incorporating plasticizers (glycerol, sorbitol), crosslinkers (glutaraldehyde, tannic acid), and fillers (nanoparticles or nanofibers). Nanoenabled delivery systems can also be designed to control the release of active ingredients (such as antimicrobials or antioxidants) into the packaged food over time, which may extend their efficacy. This article reviews the different kinds of nanocarriers available for loading active compounds into these types of packaging materials and then discusses their impact on the optical, mechanical, thermal, barrier, antioxidant, and antimicrobial properties of the packaging materials. Furthermore, it highlights the different kinds of bioactive compounds that can be incorporated into biopolymer-based packaging.

## 1. Introduction

The main factors reducing the quality, shelf life, and safety of foods are microbial spoilage, chemical reactions, and respiration, which are exacerbated by improper packaging and storage conditions. The deterioration of foods through these mechanisms increases food waste and foodborne illnesses, thereby reducing the sustainability and health of the food supply. The development and utilization of advanced packaging materials is an important strategy to overcome these problems [1]. Currently, petroleum-based packaging materials are primarily used for this purpose because of their low cost, ease of large-scale production, and excellent functional performance [2]. Nevertheless, synthetic plastics cannot be produced sustainably due to their poor biodegradability [3]. Furthermore, they cause environmental pollution and have negative effects on human health [4]. For these reasons, the design and development of eco-friendly sustainable packaging materials has received considerable attention [5]. Biopolymer-based packaging materials, which are typically constructed from proteins and/or polysaccharides extracted from animal or plant sources, are being explored as possible alternatives to petroleum-based ones [6,7]. However, these biopolymer-based materials often have limitations in terms of their functional properties. The functional performance of this kind of packaging material can often be improved by incorporating additives, such as light blockers, plasticizers, crosslinkers, diffusion blockers, antimicrobials, antioxidants, and sensors [8,9,10,11,12,13]. These additives can be used to alter the optical properties, mechanical strength, barrier properties, and stability of packaging materials, as well as to provide an indication of the quality, safety, or age of a packaged product. In some cases, the active agents incorporated into a packaging material may be designed to diffuse into the food over time so as to provide a prolonged effect, e.g., antimicrobials or antioxidants [14]. However, the direct incorporation of many of these active agents into biopolymer-based films is challenging because they are volatile, water-insoluble, chemically unstable, matrix-incompatible, or adversely impact film properties [14]. Nanoencapsulation of active agents can be used to overcome many of these problems [15,16]. In this case, the active agents are first incorporated into a well-designed nanoparticle, which is then introduced into the biopolymer-based packaging material. In this article, we review the different kinds of nanocarriers available for loading active agents and then discuss their effects on the properties of biopolymer-based packaging materials. We then highlight various applications of packaging materials containing nanoencapsulated active compounds in foods.

## 2. Overview of Nanocarriers for Active Compounds

Numerous kinds of nanocarriers are available that could potentially be used to incorporate active agents into biopolymer-based packaging materials, including microemulsions, nanoemulsions, solid lipid nanoparticles, nanostructured lipid carriers, nanoliposomes, biopolymer nanoparticles, and nanogels (Figure 1). Because nanocarriers based on proteins [17], polysaccharides [18], lipids [19], and other food-grade polymers [20,21] have been reviewed in detail elsewhere, we only give a brief overview here. For food applications, nanocarriers constructed from natural organic substances are preferred because of their lower environmental impact and toxicity, such as those fabricated from proteins, polysaccharides, phospholipids, and/or lipids [22]. In general, there are two main approaches for fabricating food-grade nanocarriers: top-down and bottom-up methods. Top-down methods involve the physical, chemical, or enzymatic disruption of macroscopic materials or large particles until they fall into the nanoscale range. Bottom-up methods typically involve the physical or chemical assembly of nanoparticles from molecules. The selection of a particular approach depends on the nature of the food product and the type of nanoparticles being produced. For example, bulk liquids can be broken down into nanoparticles by high pressure homogenization or microfluidization (top-down), whereas surfactant molecules will spontaneously assemble into nanoparticles due to the hydrophobic effect (bottom-up) [21].

Nanocarriers can be produced that exhibit a wide range of different particle characteristics, such as compositions (e.g., proteins, polysaccharides, phospholipids, lipids, and/or minerals), sizes (e.g., 1 nm to 1 µm), shapes (e.g., spherical, ellipsoid, rectangle, or fibrous), electrical charges (e.g., positive, negative, or neutral), surface hydrophobicities (e.g., polar to non-polar), interfacial thicknesses (e.g., thin to thick), surface chemistries (e.g., unreactive to reactive), physical state (e.g., solid or liquid), rheology (e.g., hard or soft), digestibility (e.g., digestible to indigestible), and biodegradability (e.g., degradable or not). Consequently, the packaging manufacturer must select ingredients and preparation methods that can be used to create the nanoparticle characteristics required to obtain specific functional performance. One of the main factors impacting the selection of a suitable nanocarrier is the nature of the active agent to be encapsulated. A few examples of different kinds of delivery systems are highlighted here. Biopolymer nanocarriers can be assembled from various kinds of proteins and polysaccharides such as casein, whey protein, lactoferrin, soy protein, zein, starch, alginate, carrageenan, and pectin [17,18]. These nanocarriers typically consist of small spherical particles that contain a network of aggregated biopolymer molecules inside. Nanogels can also be assembled from proteins and polysaccharides but they have a more porous structure inside that contains more water [23]. Nanoliposomes can be assembled from different kinds of phospholipids, such as milk, egg, soy, or sunflower lecithin [19]. These nanocarriers usually consist of one or more phospholipid bilayers assembled into concentric shells around an aqueous core. Nanoemulsions consist of emulsifier-coated fluid lipid droplets suspended in water and can be assembled from different kinds of emulsifiers and lipids [21]. Solid lipid nanoparticles (SLN) and nanostructured lipid carriers (NLC) consist of emulsifier-coated solid fat particles that may be fully or partially crystalline, respectively [24,25]. Active agents may be incorporated into these nanocarriers before or after their fabrication. Once prepared the encapsulated active agents can then be incorporated into the food packaging materials to control their functional performance [26]. The type of nanocarrier used for a specific food packaging application depends on numerous factors, including their loading capacity, encapsulation efficiency, stabilizing properties, light scattering/absorption properties, stability, interactions, and matrix compatibility. Each kind of nanocarrier has its unique advantages and disadvantages, which should be carefully considered. In the following section, we consider several active compounds that can be used to modulate the properties of packaging materials, with an emphasis on natural substances.

## 3. Active Compounds for the Production of Smart/Active Packaging Materials

The active compounds and agents that play an important role in improving the functional, mechanical, barrier, and structural properties of packaging films are presented in Figure 2.

### 3.1. Antimicrobials

Natural antimicrobials can be isolated from various kinds of plant-based materials, where they are often naturally produced by the plants as secondary metabolites to defend against microbial contamination [27]. However, many of these antimicrobials are difficult to incorporate into biodegradable packaging materials because they are volatile, chemically unstable, and/or have low solubility in water. Consequently, they need to be encapsulated prior to utilization [28]. Some of the most commonly used natural antimicrobials are essential oils and phytochemicals isolated from various kinds of plants [29]. Studies have shown that essential oils exhibit good antimicrobial activity against a broad spectrum of foodborne pathogens, which has been related to their ability to disrupt microbial cell membranes and to interfere with key biochemical pathways inside the cells [30]. As an example, thymol, eugenol, and cinnamaldehyde are essential oils that have been used in food packaging materials to control bacteria [31]. The most widely used phytochemical in biodegradable packaging materials is curcumin, which is a polyphenolic compound. Numerous studies have shown that curcumin exhibits strong antimicrobial properties when incorporated into packaging materials [32]. For antimicrobials, it is important that they can be incorporated into the packaging materials at a sufficiently high level, that they remain stable during storage, and that they can diffuse to the site of action (bacterial cell membranes) at a sufficiently high rate and level. The many other types of essential oils and phytochemicals that may be utilized as natural antimicrobials have been reviewed elsewhere [33,34,35,36].

### 3.2. Antioxidants

Many foods contain lipids or proteins that are susceptible to oxidation, which can reduce their shelf life, quality, and nutritional value [37]. Antioxidants can be incorporated into biodegradable food packaging materials to inhibit oxidation reactions [38]. There is increasing emphasis on the utilization of natural antioxidants for this purpose, especially plant-based ones. Like antimicrobials, many of these antioxidants are secondary metabolites that can be isolated from plant materials, such as essential oils and phytochemicals, which also have health effects [39]. For instance, quercetin is a phytochemical that can be isolated from onions that exhibits strong antioxidant activity when incorporated into packaging materials [10]. Similarly, many kinds of essential oils have also been shown to exhibit strong antioxidant properties when incorporated into packaging materials [38]. As an example, oregano (which contains thymol and carvacrol) has been shown to have good antioxidant properties when used in packaging applications [40]. The numerous kinds of essential oils and phytochemicals that can potentially be used as natural antioxidants have been reviewed in detail elsewhere [41,42,43,44,45].

### 3.3. Light Blockers

Light blockers are compounds that can block light, thereby protecting food components from photodegradation [46]. They may do this by scattering light waves (particles) and/or absorbing light waves (chromophores). Typically, there is a range of wavelengths where specific particles or chromophores are able to effectively block light. Many researchers are trying to increase the performance of food packaging materials by incorporating light blockers into them [47]. These light blocks are especially important for preventing UV light waves from penetrating into foods and promoting oxidative damage [48]. For example, bixin is a carotenoid found in annatto seeds (*Bixa orellana* L.), which has been used in packaging materials due to its ability to absorb light [49]. Many other natural substances exhibit the ability to absorb light at specific wavelengths, including proteins, carotenoids, and curcumin. In addition, nanoparticles can scatter light waves strongly in the UV region, which can help protect foods from photodegradation.

### 3.4. Plasticizers

Plasticizers are compounds used in biopolymer-based packaging materials to increase their flexibility and reduce their fragility [50]. Plasticizers function by reducing the intermolecular attractive forces and increasing chain mobility in the biopolymer matrix [51]. Polyols are often used as plasticizers in biopolymer films, such as glycerol, mannitol, sorbitol, and xylitol [52]. Other active compounds that have numerous hydroxyl groups can also be used as plasticizers in these films. For instance, tannins have been shown to act as plasticizers in these films due to the high number of hydroxyl groups [53].

### 3.5. Crosslinkers

The functional performance of biopolymer-based packaging materials, such as their mechanical, thermal, and barrier properties, can be improved by crosslinking the biopolymers [54]. The efficacy of crosslinking depends on the nature of the crosslinking agents and biopolymers present, particularly their type and concentration [55]. Crosslinks may involve physical interactions, such as electrostatic, hydrogen, hydrophobic, and van der Waals bonds, or covalent interactions such as disulfide bonds. A number of analytical methods can be used to characterize the bonds formed in biopolymer films, including X-ray diffraction, electrophoresis, Fourier transform infrared spectroscopy (FTIR), and nuclear magnetic resonance (NMR) spectroscopy [56]. Some natural phytochemicals are able to crosslink proteins and polysaccharides through hydrophobic or hydrogen bonding. For example, catechins have been shown to crosslink methylcellulose-based biopolymer films [57]. Crosslinking has been shown to improve the mechanical, stability, and barrier properties of biopolymer-based foods, thereby increasing their potential applications in foods [58].

### 3.6. Diffusion Blockers and Film Strengtheners

The barrier and mechanical properties of biopolymer packaging materials can be increased by adding inorganic or organic nanomaterials, such as titanium dioxide (TiO_2_), zinc oxide (ZnO), silver (Ag), nanocellulose, zein nanoparticles, starch nanoparticles, and other particulate substances [59]. These nanomaterials are usually spheroids, cuboids, or fibers. Inorganic nanoparticles are mostly used to fill the pores in the structure of biopolymer matrices. For example, TiO_2_-Ag nanoparticles were shown to fill in the pores in a gelatin matrix, which modulated the physicochemical properties of the gelatin films [60]. Organic nanoparticles are often incorporated into packaging materials to increase the strength and cohesion of the biopolymer networks. For instance, chitin nanofibers have been used to strengthen methylcellulose films [61]. Similarly, chitosan nanoparticles (CNPs) have been used to modulate the gas barrier properties, tensile strength, and thermal stability of biopolymer films [62,63]. The incorporation of silver nanoparticles into polylactic acid (PLA) films has been shown to increase biodegradation [64].

### 3.7. Sensors/Indicators

Smart sensors/indicators are being developed for use in biodegradable food packaging materials to provide information about the quality, spoilage, and safety of food products [65,66]. Natural pigments, such as anthocyanins and carotenoids, are often used for this purpose [67]. These pigments are selected because they change color in response to a specific environmental trigger, such as pH, oxygen exposure, temperature, or gas concentration. These natural pigments can often be extracted from plants and their by-products such as tomato peel, citrus fruit, potatoes, and soybean meal pulp. Anthocyanins change color in response to alterations in the pH of their environment, which are often indicative of changes in food quality or safety [46,68]. For instance, food spoilage often leads to the release of gases such as nitrogen, which changes the pH of the environment and leads to changes in the color of the pigments in the packaging materials, thus informing consumers about the status of the product [69]. As an example, barberry anthocyanins have been incorporated into biodegradable films comprised of methylcellulose and chitosan nanofibers, which were useful for monitoring changes in the freshness of meat by changing color during storage [12].

## 4. Impact of Bioactive Compounds on Packaging Properties

### 4.1. Physical Properties

The optical properties of packaging materials affect the appearance of foods, as well as the transmission of ultraviolet and visible light into the product. The incorporation of nanoparticles into foods can alter their optical properties by altering the absorption and scattering of light waves. The absorption of light depends on the absorption spectra of the different components included in the packaging material, whereas the scattering of light depends on the size, concentration, and relative refractive index of the nanoparticles and any other inhomogeneities. Consequently, the color, opacity, and light-blocking properties of packaging materials can be altered by controlling the type and concentration of chromophores and nanoparticles used. Many kinds of food-grade nanoparticles can be used to alter the light scattering properties of films, including nanoemulsion droplets, SLNs, NLCs, protein nanoparticles, chitin and cellulose nanofibers, and inorganic nanoparticles. In addition, many kinds of natural chromophores can be used to modulate the absorption properties, such as anthocyanins, carotenoids, and curcuminoids.

The thickness of packaging materials influences their optical, mechanical, and barrier properties. Various factors impact film thickness, including the film-forming method used and the nature of the forming solution used, such as its density, viscosity, surface tension, and the presence of nanoparticles. Typically, films should have thicknesses below about 0.25 mm (250 μm) for practical applications. Previous studies have shown that the incorporation of active agents usually increases film thickness, which has mainly been attributed to the increase in the total solid content of the system [70,71].

Water solubility (WS) and moisture content (MC) are important functional properties of biopolymer-based films. It is usually desirable that these films have a low solubility in water, otherwise, they may lose their integrity when exposed to humid environments or when they come into contact with moist foods [72]. Most biopolymer-based films are hydrophilic; therefore, they can absorb moisture and decompose, which limits their application [73]. To overcome this problem, hydrophobic active agents (such as lipid droplets or fat particles) are often incorporated into the film matrix to reduce the WS and MC [74]. In some applications, the ability of a film to dissolve when it comes into contact with a food surface may be useful for releasing active agents, such as antimicrobials or antioxidants [75].

Li and Yang [76] incorporated thymol nanoemulsions (0, 0.5, or 1.0% *w*/*w*) into gelatin films, which increased their thickness and reduced their water content. Similarly, Chu and Cheng [77] reported an increase in film thickness and reduction in moisture content when cinnamon oil nanoemulsions (4, 8, and 12%) were introduced into pullulan films. Behjati and Yazdanpanah [78] reported an increase in thickness, decrease in moisture content, and reduction in water solubility when vitamin D nanoemulsions were incorporated into quince seed gum films.

### 4.2. Mechanical Properties

The mechanical properties of packaging materials, such as their tensile strength (TS), elongation at break (EB), and elastic modulus (EM), play an important role in their functional performance [79]. In particular, the mechanical properties of packaging materials are important for protecting foods during storage and distribution. Several factors impact the mechanical properties of biodegradable films, including the type, number, and strength of the interactions between the biopolymer molecules [80,81]. In addition, the incorporation of certain kinds of active compounds into these films may either increase or decrease their mechanical properties depending on how they interact with the biopolymer network (Table 1).

As a specific example, the tensile strength of whey protein films has been reported to decrease from 10.8 to 3.3 MPa, the elongation at break to increase from 29.4 to 48.6%, and the elastic modulus to decrease from 2.4 to 48.6 MPa after adding 2% α-tocopherol nanoemulsion [82]. In another study, it was reported that adding α-tocopherol nanocapsules reduced the TS from around 37 to 23 MPa, and the YM from around 114 to 41 MPa in carboxymethyl cellulose films when the nanocapsule concentration was increased from 0 to 70%, while the elongation at break (EAB) significantly (*p* < 0.05) increased from around 32% to 53% [83]. These changes can at least be partially attributed to the surfactants used during nanocapsule fabrication, since the presence of surfactants can reduce intermolecular attractive forces and increase polymer mobility. Lecithin-based nanocapsules have been reported to result in softer materials with lower TS by causing discontinuities in the biopolymer films. Fattahi and Ghanbarzadeh [84] reported that incorporating a cinnamon oil nanoemulsion into a carboxymethyl cellulose matrix reduced the TS and increased the EB of the films. Incorporating essential oils probably weakened the network structure through interruption of the intermolecular interactions between the biopolymer molecules. Aziz and Almasi [85] reported that incorporating thyme extract nanoliposomes (0, 5, 10, or 15% wt.) into whey protein films altered their mechanical properties. The TS and YM values decreased from around 7.1 to 2.1 MPa and 83 to 13 MPa, respectively, as the concentration of the extract increased. This phenomenon may again be due to interruption of the biopolymer–biopolymer interactions by the nanoparticles. Conversely, Ranjbaryan and Pourfathi [14] reported that the addition of 5% cinnamon oil nanoemulsion had no significant effect on the EB of sodium caseinate films, but increased their TS and EM.

### 4.3. Microstructural Properties

The microstructure of biodegradable films influences their functional properties, such as their optical, mechanical, and barrier properties. Microscopic techniques, such as scanning electron microscopy (SEM), confocal laser scanning microscopy (CLSM), and atomic force microscopy (AFM), are commonly used to evaluate the microstructures of films. The incorporation of certain kinds of active agents can affect film microstructure, owing to their ability to form covalent and non-covalent interactions with the reactive groups on the biopolymer chains [86,87]. For example, Ghasempour and Khodaivandi [88] showed that incorporating betanin nanoliposomes into films made from Persian gum and whey protein increased the roughness of their microstructure. However, Najafi and Kahn [89] showed that saffron nanoliposomes could be incorporated into pullulan films without impacting their smooth homogenous structure. Li and Yang [76] also showed that uniform films could be produced by incorporating thymol nanoemulsions (0, 0.5, or 1.0% *w*/*w*) into gelatin films. Conversely, the incorporation of α-tocopherol nanocapsules into carboxymethyl cellulose led to the formation of a heterogenous structure with high porosity and many cracks [83]. Kong, Wang [90] showed that incorporation of carvacrol nanoemulsions into corn starch-poly(vinyl alcohol) (PVA) films led to a dense uniform microstructure. Chen, Ma [87] showed that eugenol-loaded gelatin nanofibers could be deposited on the surfaces of poly (lactic acid) films, which led to the formation of smooth uniform surfaces. Overall, these studies show that different kinds of nanoparticles have different effects on different biopolymer films. Consequently, it is important to examine these effects for specific combinations of nanoparticles and films.

### 4.4. Thermal Stability

The resistance of films to thermal degradation is important to maintain the integrity and performance of the packaging material through processing, storage, distribution, and utilization, as well as determining its suitability for disposal using thermal combustion methods. Mirzaei-Mohkam and Garavand [83] evaluated the thermal properties of carboxymethyl cellulose films containing vitamin E nanocapsules. Different scanning calorimetry (DSC) showed that incorporation of the nanocapsules slightly reduced the melting temperature of the films, which was attributed to their effect on the interactions of the carboxymethyl cellulose chains. Najafi and Kahn [89] reported that the addition of nanoliposomes reduced the glass transition temperature (T_g_) of pullulan films from around 100 to 78 °C, which was attributed to a plasticizing effect by lecithin. In a recent study, pH-sensitive films were prepared by incorporating curcumin nanocapsules (0.03% *w*/*v*) into soy protein-cellulose nanocrystals films [91]. Thermogravimetric analysis showed that the maximum degradation temperature of the films increased after the addition of the curcumin nanocapsules, which was mainly attributed to the ability of the curcumin to crosslink the biopolymers in the films. de Carvalho and Noronha [92] showed that loading α-tocopherol SLNs into PVA films increased their thermal stability and reduced their crystallinity. These studies show that the thermal stability of biopolymer-based films can be modulated by incorporating different kinds of nanoparticles.

### 4.5. Barrier Properties

The shelf life, safety, and quality of foods are influenced by the transfer of water vapor and other gases between the internal and external environments of the packaging materials, as well as by the diffusion of other substances through the film, such as liquid water, oils, and nanoparticles. Controlling the flow of oxygen is often important in biopolymer packaging materials because it can influence the oxidation of foods [93]. Controlling the flow of water vapor is important because it influences the texture and water activity of foods, which in turn influences chemical reactions and microbial growth. The permeability of biopolymer films depends on their porosity, thickness, and rheology. Hence, it can be controlled by adding different kinds of active agents into the films, such as nanoparticles, nanofibers, plasticizers, or crosslinking agents [94]. These substances may either increase or decrease the permeability of the films depending on their effects on biopolymer interactions and porosity (Table 1) [95].

The water vapor permeability (WVP) of edible films was shown to increase from 12.1 to 19.4 × 10^−10^ g s^−1^ m^−1^ Pa^−1^ when 5 to 15% rutin nanoemulsions were added but then decreased to 18.7 × 10^−10^ g s^−1^ m^−1^ Pa^−1^ when 20% of these nanoemulsions was added [6]. Presumably, the nanoemulsions acted as a plasticizer of the biopolymer molecules in the films, which facilitated the transfer of water vapor through the films. Similarly, Ji and Wu [96] and Sun and Wang [97] reported that the addition of cinnamaldehyde and lavender oil nanoemulsions increased the WVP of gelatin films, which may have been because the essential oil reduced the number of hydrogen bonds between the protein molecules. In contrast, the addition of 3% caraway seed extract-loaded nanoliposomes was reported to reduce the WVP of nanochitosan-based films from 14.2 × 10^–12^ g/m·s·Pa to 11.9 × 10^–12^ g/m·s·Pa [98]. Similarly, Liu and Dang [99] showed that incorporating curcumin-loaded Pickering emulsions into corn starch-PVA films decreased their permeability to oxygen and water vapor. Sun and Li [100] also reported a decrease in water vapor and oxygen permeabilities when cinnamon oil Pickering emulsions were added of modified starch films. In another study, Bi and Qin [101] showed that adding a luteolin nanoemulsion to a chitosan film reduced the WVP and oxygen permeability [83]. Most of these effects can be attributed to the presence of hydrophobic particles within the biopolymer films that made the diffusion pathway for oxygen and water molecules more tortuous.

### 4.6. Antioxidant Properties

Lipid and protein oxidation are important chemical reactions that result in quality loss during food processing, storage, and distribution. Natural antioxidants and scavengers can be incorporated into packaging materials to inhibit or prevent these reactions. The susceptibility of packaged foods to oxidation is often assessed using in vitro assays, such as the DPPH, ABTS, and FRAP assays [102].

Mirzaei-Mohkam and Garavand [103] showed that adding α-tocopherol nanocapsules to carboxymethyl cellulose (CMC) increased their radical scavenging activities in a dose-dependent manner. Various other kinds of antimicrobial nanoparticles have also been used to increase the antioxidant activity of packaging films, including carvacrol nanoemulsions in corn starch/PVA films [90], rutin nanoemulsions in gelatin films [6], α-tocopherol nanoemulsions in whey protein films [82], clove oil nanoemulsions in pullulan-gelatin films [104], and clove oil Pickering emulsions in gelatin/agar films [105]. The observed increase in antioxidant activity has mainly been attributed to the presence of numerous hydroxyl groups in the encapsulated active agents, which act to donate electrons to reactive free radicals during oxidation, thus converting them into more stable non-reactive species, which breaks the free radical chain reaction [106,107].

### 4.7. Antimicrobial Properties

Some active agents exhibit strong antimicrobial properties and can therefore be incorporated into packaging materials to inhibit the growth of spoilage and pathogenic microorganisms, thereby improving food quality and safety, and reducing food waste [108]. As mentioned earlier, many natural antimicrobials (like essential oils and phytochemicals) are strongly hydrophobic substances and must therefore be incorporated into nanocarriers before they can be introduced into biopolymer-based films [109]. Encapsulation improves the dispersibility, matrix compatibility, and stability of the antimicrobials, as well as allows control of their release profiles [110]. Various kinds of natural active agents have been investigated for their ability to increase the antimicrobial activity of biopolymer films. For instance, Kong and Wang [90] showed that the antifungal properties of corn starch-PVA films were increased by incorporating carvacrol nanoemulsions. Similarly, Chen and Ma [87] found that the incorporation of eugenol-loaded gelatin nanofibers into biopolymer films increased their ability to inhibit *E. coli* and *S. aureus*. In another study, Beikzadeh and Akbarinejad [111] reported that lemon myrtle essential oil-loaded cellulose acetate nanofibers increased the antibacterial properties of biopolymer films against *B. cereus* and *E. coli*. The incorporation of thyme oil nanoemulsions into chitosan films enhanced their antimicrobial activity against *E. coli* and *Bacillus subtilis* [112]. Lee and Garcia [70] reported that the introduction of oregano oil nanoemulsions into hydroxypropyl methylcellulose (HPMC) films improved their antimicrobial activity against *S. aureus*, *B. cereus*, *L. monocytogenes*, *E. coli*, *S. typhimurium*, *P. aeruginosa*, and *V. parahaemolyticus*. These antimicrobial effects are mainly attributed to the ability of the active agents to diffuse out of the nanoparticles, through the films, and to the surfaces of the microorganisms. Once there, they damage the cell walls and interfere with critical biochemical pathways, thereby deactivating the microorganisms.

**Table 1 polymers-13-04399-t001:** Impact of bioactive compounds on film packaging material characteristics.

Polymer	Bioactive Compound Type	Bioactive Compound (wt%)	Tensile Strength (MPa)	Elongation at Break (%)	Water Vapor Permeability	Antimicrobial	Antioxidant	Thermal	Microstructural	Reference
Cassava starch	Lycopene	0%	3.09 ± 0.10	134.59 ± 2.69	0.36 ± 0.05	-	-	residual mass = 6%	Uniform and compact	[113]
		2%	2.81 ± 0.06	233.13 ± 1.07	0.57 ± 0.02			-	Porous and Non-uniform	
		5%	2.92 ± 0.07	190.73 ± 0.96	0.55 ± 0.04			residual mass = 7%		
		8%	2.66 ± 0.04	166.03 ± 0.93	0.55 ± 0.03			-		
Cassava starch	Bixin	0%	12.13 ± 0.95	6.05 ± 0.72	0.207 ± 0.014	-	-	High thermal stability at least up to 270 °C	Compact and uniform structures	[114]
		2%	14.40 ± 1.69	2.19 ± 0.35	0.202 ± 0.008					
		5%	8.95 ± 1.32	15.55 ± 1.14	0.216 ± 0.007					
		8%	2.06 ± 0.34	28.57 ± 3.44	0.243 ± 0.010				Cracks surface	
		10%	1.94 ± 0.37	34.34 ± 3.40	0.273 ± 0.018					
Cassava starch	β-carotene	0%	3.09 ± 0.10	134.59 ± 2.69	0.36 ± 0.05	-	-	No effect on thermal stability	Non-uniform structure	[115]
		2%	2.74 ± 0.19	237.81 ± 7.49	0.45 ± 0.02				-	
		5%	2.56 ± 0.15	311.82 ± 6.73	0.44 ± 0.05				Smooth surface with pores	
		8%	2.63 ± 0.18	319.74 ± 3.35	0.44 ± 0.03				Heterogeneous and cracks structure	
HPMC	Nisin	0%	59.0 ± 6.8	6.0 ± 3.3	0.77 ± 0.03	Antimicrobial activity against *Listeria monocytogenes*	-	-	Smooth surface	[116]
		100%	37.0 ± 2.5	2.6 ± 0.7	0.95 ± 0.10				Non-uniform surface with dome-shaped zones and holes	
Pullulan	Lysozyme	0%	35.0 ± 4.4	6.63 ± 1.11	-	Antimicrobial activity against *Staphylococcus aureus*	High antioxidant activity (77%) for 15% LNFs	High thermal stability at least up to 225 °C	Homogeneous, smooth, compact surface	[117]
		1%	33.2 ± 3.7	2.57 ± 0.36						
		3%	35.6 ± 2.2	2.24 ± 0.27						
		5%	37.6 ± 2.2	1.84 ± 0.29						
		10%	34.1 ± 1.0	1.64 ± 0.61						
		15%	31.3 ± 2.3	1.34 ± 0.10						
Chitosan	Epigallocatechin gallate	0%	6.44 ± 0.28	22.5 ± 4.3	-	-	Higher DPPH scavenging activity	-	Smooth	[118]
		2.5%,	10.4 ± 3.2	24.1 ± 4.6					Rough and uneven surface	
		4.5%	19.2 ± 1.2	20.6 ± 3.5						
		6.0%	18.10 ± 4.10	3.9 ± 2.6						
Chitosan	Cinnamaldehyde	0%	98.26 ± 5.69	4.16 ± 0.47	1.42 ± 0.29	Better antifungal than antibacterial activity	-	-	Bubble-like surface	[81]
		0.1%	62.29 ± 3.47	16.1 ± 2.6	1.31 ± 0.36				Uniform and smooth	
		0.2%	51.78 ± 4.70	24.5 ± 0.6	1.15 ± 0.09					
		0.4%	44.90 ± 4.11	12.2 ± 3.5	1.69 ± 0.05					
		0.6%	37.42 ± 4.02	14.5 ± 2.9	1.74 ± 0.14					
		0.8%	38.84 ± 4.74	14.6 ± 2.5	2.01 ± 0.27					
		1.0%	29.57 ± 4.21	11.4 ± 2.6	2.24 ± 0.17					
		1.5%	17.44 ± 3.48	14.9 ± 3.7	3.33 ± 0.47					
		2%	7.57 ± 1.34	12.6 ± 2.2	3.91 ± 0.59					
Soy protein isolate	Cinnamaldehyde/ Carvacrol	0	2.61 ± 0.54	172 ± 46	2.89 ± 0.24	-	-	-	-	[119]
		Carvacrol	1.97 ± 0.11	418 ± 37	2.79 ± 0.28					
		Cinnamaldehyde	2.52 ± 0.21	374 ± 50	2.83 ± 0.09					
Low pectin	Cinnamaldehyde	0	5.36 ± 0.42	246 ± 23	3.10 ± 0.10	Good antimicrobial activity	-	-	-	[120]
		7	6.34 ± 0.71	174 ± 40	2.92 ± 0.09					
		12	5.99 ± 0.14	139 ± 29	2.15 ± 0.10					
		16	6.53 ± 0.68	146 ± 17	2.90 ± 0.10					
High pectin	Cinnamaldehyde	0	4.84 ± 0.29	192 ± 39	3.26 ± 0.02		-	-	-	
		7	7.70 ± 1.17	155 ± 30	3.02 ± 0.02					
		12	7.62 ± 0.2	170 ± 36	2.70 ± 0.02					
		16	8.36 ± 0.15	180 ± 14	2.95 ± 0.02					
Quinoa protein/chitosan	Thymol	0	4.4 ± 0.7	116 ± 22	0.35 ± 0.05	-	-	-	Homogeneous	[121]
		10%	2.9 ± 0.5	98 ± 11	0.40 ± 0.04				Porous and heterogeneous surface	

## 5. Application of Active-Loaded Packaging Materials in Foods

In this section, we give several examples of studies where active-loaded packaging materials have proved useful for improving the quality, safety, or shelf-life of foods. The results of many of these studies are also summarized in Table 2 for convenience.

### 5.1. Seafood

Seafood products are highly perishable because they are rich in nutrients and moisture that microorganisms can use to grow and multiply. Moreover, they often contain sensitive components (such as polyunsaturated lipids and proteins) that are prone to chemical degradation. Microorganisms generate several degradation products when they grow on seafood, including dimethylamine (DMA), trimethylamine (TMA), and ammonia. Similarly, oxidation products, such as peroxides, conjugated dienes, TBARS, and aldehydes are produced when polyunsaturated lipids degrade. These degradation products can therefore be measured to monitor the quality and freshness of these products.

Nanoparticles loaded with active agents can be incorporated into biodegradable packaging materials to help protect seafood by inhibiting microbial growth and chemical degradation. For instance, Sharifimehr and Soltanizadeh [122] incorporated eugenol nanoemulsions into *aloe vera* coatings on pink shrimp and showed that the formation of lipid oxidation products, as well as drip losses and color changes, were reduced. Nazari and Majdi [123] showed that incorporation of cinnamon oil nanophytosomes into PVA coatings increased their antimicrobial activity against *P. aeruginosa* on shrimp. Homayonpour and Jalali [124] showed that incorporation of cumin oil nanoliposomes into nanochitosan-based coatings improved the quality and shelf life of sardine fillets during storage under refrigerator conditions. Xiao and Liu [91] created pH-sensitive films by incorporating curcumin nanocapsules into soy protein-cellulose nanocrystal films and showed they could be used to monitor changes in the freshness of shrimp during refrigerated storage. The color of the films turned from bright yellow to reddish-brown, which was attributed to the deterioration of the shrimp.

### 5.2. Meat

Like seafood, meat is also rich in nutrients and moisture, which makes it susceptible to microbial and chemical spoilage. Poultry meat is easily spoiled by various kinds of bacteria and yeast during cold storage [125]. The incorporation of *Trachyspermum ammi* oil nanoemulsions into alginate edible coatings was shown to inhibit the growth of *Listeria monocytogenes* on turkey fillets during refrigerator conditions [126]. Kamkar and Molaee-Aghaee [9] showed that adding *garlic* oil nanoemulsions to chitosan films inhibited the growth of *aerobic*, *psychrotrophic*, and *coliform* bacteria on the chicken breast during refrigerated storage, which improve their quality attributes and shelf life.

Fresh meat is also easily spoiled by microbial growth and chemical reactions [127]. Xavier and Sganzerla [128] showed that incorporation of *cinnamodendron dinisii* oil-loaded zein nanoparticles into chitosan films increased the storage stability of refrigerated ground beef, which was attributed to the antioxidant and antimicrobial activity of the encapsulated essential oil. In another study, Amjadi and Nazari [129] incorporated chitosan nanofibers, ZnO nanoparticles, and betanin nanoliposomes into gelatin films, which improved the quality and extended the shelf life of refrigerated beef meat. Xiong and Li [130] reported that incorporating oregano oil- and resveratrol-loaded nanoemulsions into pectin coatings improved the quality and shelf life of pork loin during refrigerated storage. In another study, Zhang and Liang [131] reported that incorporating tarragon oil-loaded nanoparticles into chitosan-gelatin coatings applied to pork slices improved their quality and shelf life. Overall, these effects can mainly be attributed to the antimicrobial and antioxidant properties of the additives in the films.

### 5.3. Cheese

Cheese is another nutrient-rich food that is susceptible to microbial and chemical spoilage [132]. Al-Moghazy and El-sayed [133] assessed the effect of thyme oil-loaded nanoliposomes incorporated into chitosan coatings on the quality of Karish cheese. The coatings did not affect the initial quality attributes of the cheese, but they did extend its shelf life by inhibiting the growth of spoilage microorganisms. Incorporating nisin-loaded nanoparticles within polyethylene oxide nanofibers was also shown to inhibit *L. monocytogenes* growth on cheese [134].

### 5.4. Bread

Lei and Wang [135] prepared carboxymethyl cellulose films containing carvacrol nanoemulsions. These films were shown to extend the shelf life of wheat bread stored at 25 °C, which was mainly attributed to the antioxidant and antimicrobial activity of the essential oil. Indeed, these films inhibited the growth of *E. coli* and *S. aureus*, as well as aerobic bacteria, molds, and yeast. In another study, Otoni and Pontes [136] incorporated clove bud and *oregano* oil nanoemulsions into methylcellulose films, which were then shown to increase the shelf life of sliced bread during storage by reducing the growth of yeasts and molds.

### 5.5. Fruit and Vegetables

The perishable nature of fruits and vegetables results in a short life and loss of quality. The main causes of spoilage are transpiration and respiration, ethylene production, and fungal growth [137]. β-carotene nanoparticles have been incorporated into xanthan gum coatings that were applied to fresh-cut melon [138], which improved the firmness and juiciness of the melon. Ansarifar and Moradinezhad [139] applied thyme oil-loaded zein nanofibers to strawberries and showed that they inhibited the growth of fungi and yeast, as well as reduced lipid oxidation. Again, these results can mainly be attributed to the antioxidant and antimicrobial properties of the additives in the films.

**Table 2 polymers-13-04399-t002:** Application of bioactive packaging films loaded with bioactive agents on food samples.

Active Agent	Matrix	Natural Compounds	Nanocarrier	Food Model	Condition Storage	Ref.
Phenolic compounds	*Aloe vera*	*Eugenol* essential oil (EO)	Nanoemulsion	Shrimp	-	[122]
	*Aloe vera*	*Eugenol* EO	Nanoemulsion	Shrimp	-	[140]
	Gelatin-Carrageenan	Curcumin Gallic acid Quercetin	Nanoemulsion	Raw broiler meat	20 days at 4 °C	[141]
	Alginate-CMC	Vanillin Ascorbic acid	Nanoemulsion	Fresh cut kiwi slices	7 days at 5 ± 1 °C	[142]
	Pectin	Resveratrol	Nanoemulsion	Pork	15 days at 4 °C	[130]
	Chitosan	*Thymol* or *thyme* EO	Nanoemulsion	Pork	12 days at 4 °C	[143]
Essential oil	Chitosan	Cinnamodendron dinisii Schwanke	Nanoparticle	Ground beef	12 days at 6 ± 2 °C	[128]
	Chitosan	Paulownia Tomentosa	Nanoparticle	Pork chop	16 days at 4 °C	[144]
	Chitosan-gelatin	Tarragon	Nanoparticle	Pork	16 days at 4 °C	[131]
	Corn starch	*Zataria multiflora* EO	Nanoemulsion	Chicken meat		
	PVA	Cinnamon	Nanophytosome	Shrimp	7 days at 4 °C	[123]
	Pullulan	Cinnamon	Nanoemulsion	Strawberries	6 days at 20 ± 2 °C	[145]
	Nanochitosan	*Cuminum cyminum* EO	Nanoliposome	Sardine	16 days at 4 °C	[124]
	Chitosan	*Ferulago angulata EO*	Nanoemulsion	Rainbow trout fillets	16 days at 4 °C	
	Gelatin/Hydroxypropyl	Mustard	Nanoemulsion	Turkey	20 days at 4 ± 1 °C	[146]
	Chitosan	Garlic	Nanoliposome	Chicken breast fillets	at 4 °C	[9]
	Alginate	*Oregano* EO	Nanoemulsion	Tomatoes	14 days at 14 days at 24 ± 1 °C	[147]
	Alginate	Trachyspermum ammi	Nanoemulsion	Turkey fillets	12 days at 4 ± 1 °C	[126]
Carotenoid	Xanthan gum	β-carotene	Nanocapsule	Fresh cut melon	-	[138]
Peptide	Soy Protein Isolate	*Star anise* essential oil Polylysine Nisin	Nanoemulsion	Yao meat	20 days at 4 °C	[147]
	Poly (ethylene oxide)	Nisin	Nanoparticle	Cheese	-	[134]
Others	Gelatin	Betanin	Nanoliposome	Beef	16 days at 4 °C	[129]

## 6. Conclusions and Future Directions

A broad range of nanoparticle-based delivery systems are available to incorporate active agents into biodegradable packaging materials, including nanoemulsions, nanoliposomes, SLNs, NLCs, biopolymer nanoparticles, and nanogels. These active-loaded nanoparticles can be used to improve the physicochemical and functional attributes of the packaging materials, including their optical properties, mechanical strength, barrier properties, and stability, as well as to provide indications of the quality, safety, or age of the packaged product. The selection of an appropriate nanoenabled delivery system depends on the nature of the active agent, food product, and film matrix. Many different kinds of food-grade active ingredients can be incorporated into biodegradable packaging materials, including phytochemicals that exhibit antioxidant, antimicrobial, light blocking, film strengthening, and indicator properties. The delivery system used should be designed to improve the dispersibility, matrix compatibility, stability, and efficacy of these active agents, as well as to improve or extend the functional performance of the packaging material. These delivery systems can also be designed to control the release of the encapsulated active agents from the film matrix during storage, which may be advantageous for some applications. For instance, extended release of antioxidants or antimicrobial agents from the packaging materials into the food product may extend the shelf life of some foods. In the future, it will be important to test these fortified packaging materials under real-life conditions, as well as to establish the economic feasibility of their mass production. In addition, it will be important to test their safety and environmental impact.

## Figures and Tables

**Figure 1 polymers-13-04399-f001:**
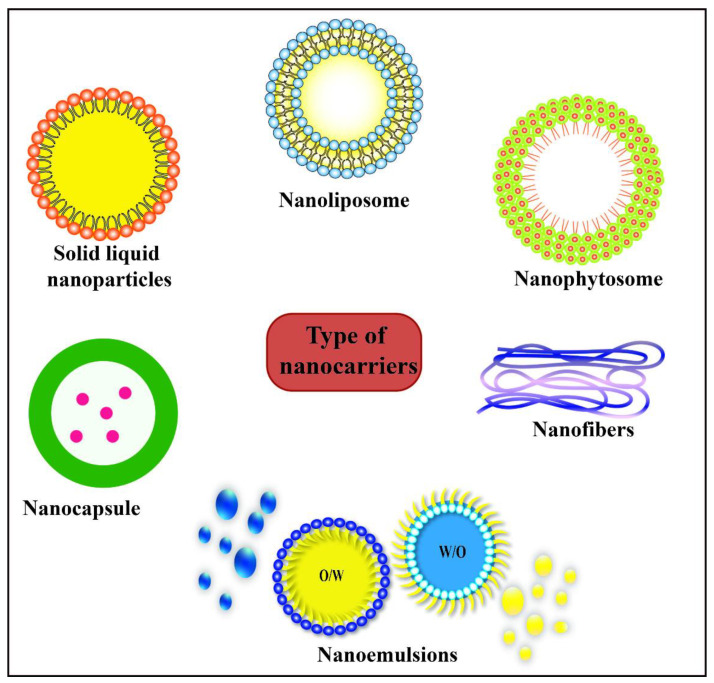
Types of nanocarriers for use in food packaging material.

**Figure 2 polymers-13-04399-f002:**
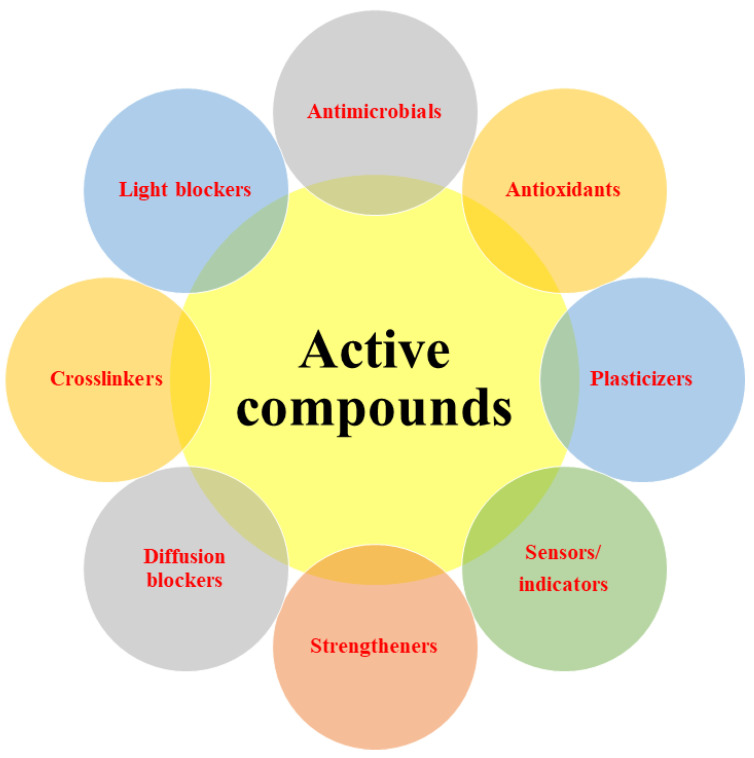
Active compounds for the production of smart/active packaging materials.

## Data Availability

Not applicable.

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
