# Peer review of "Application of Nanotechnology to Improve the Performance of Biodegradable Biopolymer-Based Packaging Materials"

_polymers, 2021, doi:10.3390/polym13244399_

Round 1

Reviewer 1 Report

Dear author,

Here are my comments for your paper:

  1. Please state the novelty of the work with respect to existent literature. The added values must be shown. Most of the aspects shown in the paper are known.
  2. Which is your personal contribution to the paper in terms of research?

Author Response

Author's Reply to the Review Report

Reviewer 2 Report

In general, the paper is interesting. Its main concept is appropriate. Paper may be considered for publication but some revisions are required. All suggestions are given in more detail below.

  • abstract of the paper should focus more on conclusions drawn based on performed literature review, and not present only general sentences concerning the background of the research topic undertaken by Authors;
  • notation [17-24] is not appropriate – Authors should divide such reference range into smaller ones and discuss in more detail information from the mentioned publications;
  • in the first sentence below Figure 1. Authors provide information concerning nanocarriers and report on their size within the range 10 – 1000 nm. This issue should be explained because, in general, term “nano” refers to the range 1 – 100 nm;
  • polymer notation should be corrected in the whole paper, e.g. instead of “polyethylene oxide” Authors should write “poly(ethylene oxide)”;
  • Figure 2.: it should be moved at the beginning of Section 3.; by the way it looks better when a figure is within the text, and not directly under or below the name of section/subsection;
  • Table 1.: on page 12 the table overlaps the text therefore it is difficult to read its content – it should be improved;
  • all abbreviations used in the paper should be explained when they are used for the first time; e.g. abbreviations “carv” and “cinn” in Table 1. or “SPI” and “EO” in Table 2.;
  • all Latin names should be italicized (e.g. Aloe vera) – it should be checked in the entire paper,
  • Section “Conclusions” should be significantly extended;
  • Section “References” should be improved and its notation should be consistent, e.g. now some references contain the whole journal names and some contain their abbreviations.

Author Response

Author's Reply to the Review Report

Reviewer 3 Report

Khezerlou et al. aim with this article to review the different types of nanoparticles available for loading active compounds, and then to discuss their impact on the optical, mechanical, thermal, barrier, antioxidant, and antimicrobial properties of the packaging materials. Overall, this paper is well organized and written. Some recommendations should be addressed to improve the quality of the review.

Table 1: The authors refer to table 1 on page 8 and 10. However, the data presented are not discussed, or even mentioned in the main text. Authors should discuss on the data presented.

Page 15-17: The authors present some examples of applications of active-loaded packaging materials in foods. However, most of these examples are related to the antioxidant and antimicrobial activities of the compound, and as the authors presented, active compounds can also act as plasticizers, sensors, strengtheners, diffusion blockers, crosslinkers or light blockers. Thus, the authors should present at least one or two examples of each of these properties.

Table 2: Authors should specify the type of nanoparticles instead of just presenting “nanoparticle” and present the main conclusions of these studies.

Conclusions and future directions: Authors should discuss on the current limitations of this field and further develop the future directions.

Some minor concerns:

  • Page 1: The keywords “Packaging materials” and “Nanotechnology” already appear in the title. Please replace these words with others in order to increase the visibility of the article.
  • Page 17: What do the authors mean by “real food samples”?
  • There are minor errors throughout the text that should be corrected.
  • Avoid abbreviations when used once.

Author Response

Author's Reply to the Review Report

Round 2

Reviewer 1 Report

Dear authors,

The paper has been improved and it could be accepted.

Reviewer 3 Report

The authors answered satisfactorily to the comments.